# Global Pandemic Preparedness: Optimizing Our Capabilities and the Influenza Experience

**DOI:** 10.3390/vaccines10040589

**Published:** 2022-04-12

**Authors:** Steven Rockman, Beverly Taylor, John W. McCauley, Ian G. Barr, Ray Longstaff, Ranbir Bahra

**Affiliations:** 1Seqirus Ltd., Parkville, VIC 3052, Australia; 2Department of Immunology and Microbiology, University of Melbourne, Parkville, VIC 3052, Australia; 3Seqirus Ltd., Maidenhead SL6 8AA, UK; beverly.taylor@seqirus.com (B.T.); ray.longstaff@seqirus.com (R.L.); ranbir.bahra@seqirus.com (R.B.); 4The Francis Crick Institute, London NW1 1AT, UK; john.mccauley@crick.ac.uk; 5WHO Collaborating Centre for Reference and Research on Influenza, Melbourne, VIC 3000, Australia; ian.barr@influenzacentre.org

**Keywords:** influenza, pandemic, vaccines, vaccine manufacturing

## Abstract

The coronavirus disease 2019 (COVID-19) pandemic has prompted rapid investigation and deployment of vaccine platforms never before used to combat human disease. The severe impact on the health system and the high economic cost of non-pharmaceutical interventions, such as lockdowns and international border closures employed to mitigate the spread of COVID-19 prior to the arrival of effective vaccines, have led to calls for development and deployment of novel vaccine technologies as part of a “100-day response ambition” for the next pandemic. Prior to COVID-19, all of the pandemics (excluding HIV) in the past century have been due to influenza viruses, and influenza remains one of the most likely future pandemic threats along with new coronaviruses. New and emerging vaccine platforms are likely to play an important role in combatting the next pandemic. However, the existing well-established, proven platforms for seasonal and pandemic influenza manufacturing will also continue to be utilized to rapidly address the next influenza threat. The field of influenza vaccine manufacturing has a long history of successes, including approval of vaccines within approximately 100 days after WHO declaration of the A(H1N1) 2009 influenza pandemic. Moreover, many advances in vaccine science and manufacturing capabilities have been made in the past decade to optimize a rapid and timely response should a new influenza pandemic threat emerge.

## 1. Introduction

In the past century, five global pandemics—four influenza and the current coronavirus disease 2019 (COVID-19) pandemic—have caused tens of millions of deaths worldwide as well as massive disruptions to economies, global progress, and people’s lives. Given the major burden on society and individuals, international organizations, governments, academia, and the pharmaceutical industry have invested and will continue to invest heavily in pandemic research and preparedness in cooperation with public health authorities. This manuscript explores the current established influenza pandemic infrastructure which is commercially sustainable and, at this time, represents the most effective response to an emerging novel influenza virus.

The COVID-19 pandemic has expedited development of vaccines against severe acute respiratory syndrome coronavirus 2 (SARS-CoV-2). Globally, over 140 COVID-19 vaccines are in clinical development and nearly 200 in preclinical development, encompassing 11 different vaccine platforms [1]. As of February 2022, 16 different vaccines have been authorized by at least one regulatory authority, and 10 of these have received emergency use listing from the WHO [1,2]. The speed with which these vaccines against a novel pathogen have reached the market has prompted both celebration and hesitancy; however, the research behind them spans several decades. For example, the first mRNA vaccines and first adenovirus vector vaccines were investigated in the 1980s [3,4,5].

These novel COVID-19 vaccines have proven invaluable in efforts to counter the pandemic, and the ongoing effort to control the virus will continue to rely heavily on these and other novel platforms as they are approved [6]. The COVID-19 vaccine successes have generated interest in emerging vaccine technologies for potential applications ranging from: zoonoses such as avian and swine variant influenza viruses, endemic infections such as Zika, Dengue, Chikungunya, and non-infectious diseases such as cancer [7]. These platforms are also being explored as vaccines against seasonal influenza and respiratory syncytial virus (RSV) and as combination vaccines that could protect against influenza, RSV, and SARS-CoV-2 in a single vaccine [8].

However, influenza remains one of the most likely causes of future pandemics. The World Health Organization (WHO) Global Influenza Surveillance and Response System (GISRS) and other agencies worldwide maintain global surveillance to detect and monitor influenza outbreaks and emerging influenza viruses of pandemic potential (IVPPs) [9,10]. These evaluations are made available through regular reports such as the WHO monthly risk assessment of influenza at the human-animal interface, the CDC Influenza Risk Assessment Tool (IRAT), and the WHO Tool for Influenza Pandemic Risk Assessment (TIPRA; WHO) [11,12,13]. This ongoing threat presented by potential pandemic viruses, especially influenza, is well documented in government risk assessments such as the United Kingdom National Risk Register [10]. The WHO Pandemic Influenza Preparedness (PIP) Framework was adopted in 2011 and is designed to improve preparedness and to facilitate responses in the event of an influenza pandemic through the co-ordination of activities between governments, industry, and other stakeholders. Of note, the framework places the sharing of PIP biological material and the sharing of benefits on an equal footing. Key elements include the rapid sharing of IVPPs by Member States and increased access of developing countries to vaccines, antivirals and other supplies [14]. However, the PIP framework has yet to be utilized during an influenza pandemic, and it is likely that the committed capacity for sharing vaccines, antiviral medicines, and diagnostics is likely to fall short of what would be needed for a global influenza pandemic response. Much of the existing infrastructure developed for influenza was utilized for the SARS-CoV-2 pandemic, including surveillance and detection by WHO GISRS and a SARS-CoV-2 sequence database established on the Global Initiative on Sharing Avian Influenza Data (GISAID) [15].

Novel vaccines technologies, which some claim can be developed faster than existing influenza vaccines, offer great hope for vaccines to counter the next pandemic. However, these new technologies are still being investigated and developed for influenza—a virus more mutable than SARS-CoV-2 with a wide range of pathogenetic traits [3,16]. In response to the last influenza pandemic, the existing influenza vaccine infrastructure including international and national bodies, industry and academics collaborated to investigate, design, and begin producing vaccines based on the novel strain of swine-lineage A(H1N1) as soon as it emerged, to enable a rapid response as soon as WHO declared a pandemic in June 2009 [17,18]. Thus, established platforms continue to remain an essential part of producing and distributing considerable quantities of influenza pandemic vaccines in a timely manner. Success of future endeavours will depend on the sound knowledge of influenza viruses and continual improvement of response capability through the ongoing study of the characteristics of subtypes, good global surveillance, established supply chains for raw materials and components, well-established (yet continually improving) and scalable manufacturing processes, optimizing vaccine virus yields, accelerated regulatory pathways, and advanced purchase agreements to help determine demand and support preparedness and sustainability (e.g., by maintaining stockpiles of key materials). This well-defined network of activities, collaboration, and years of experience in influenza mitigation will play a key role in shaping future pandemic preparedness. In addition, the lessons learned from the COVID-19 vaccine will help to improve the timeliness and global coverage of pandemic influenza vaccines.

## 2. The 100-Day Pandemic Response Ambition

The challenges encountered during the COVID-19 pandemic have led to the development of ambitious, forward-thinking strategies, including the “100-day response” and the Influenza Vaccine Roadmap (IVR) [19,20]. These approaches challenge the pandemic preparedness community to capitalize on innovations in science and manufacturing to create a more timely, efficient, and effective response before the next pandemic. For example, the Pandemic Preparedness Partnership was set up to advise the UK G7 Presidency on how to develop and deploy safe, effective diagnostics, therapeutics, and vaccines within the first 100 days of a pandemic. This G7 initiative, working with the G20 and wider international partners, the WHO, the life sciences and biotechnology sectors, and international organizations has laid out a set of recommendations to galvanize collective international action. The recommendations include global agreement on priorities for targeted virus families; further development of novel “programmable” vaccine and therapeutic platform technologies, including strengthening global manufacturing capacity, improve international coordination on clinical trials and harmonized/streamlined regulatory processes, and embedding a mission-driven approach to drive collaboration, investment and public-private partnerships [21,22].

The IVR is a strategic plan published in September 2021 and developed by the Center for Infectious Disease Research and Policy (CIDRAP) with the support of the Wellcome Trust [19]. CIDRAP brought together a large group of experts with the aim of generating a plan for “improvements in strain-specific seasonal vaccines, as well as transformational changes in vaccine technology designed to induce broad, durable protection against seasonal and pandemic influenza viruses” [19].

The United States government also announced a set of pandemic preparedness goals in September 2021. The primary goal was to “rapidly make effective vaccines against any human virus family within 100 days after a pandemic threat appears” [23]. Moreover, enough fully tested vaccines doses should be available to vaccinate the US population against the pandemic threat within 130 days and the global population within 200 days. Additional vaccine-related goals are to simplify vaccine distribution by, for example, eliminating the need for cold storage and simplifying administration by developing more non-injectable vaccines. To achieve these ambitious goals, the US government pledged to support further development of nucleic acid and recombinant viral vector technologies to speed vaccine production [23]. While these goals are commendable, the recent experience of COVID-19 and the closure of borders, disruption to international travel and supply chains, and vaccine sovereignty highlights the requirement for further planning and development in these areas. The 100-day goals represent the optimal pandemic response, and responding within that timeframe would require global agreement on pandemic triggers, vaccine development stages, regulatory mechanisms, manufacturing steps, quality assurance, and most importantly, the dissemination and delivery of vaccines in quantities large enough to equitably meet global public health needs. In addition, some important steps, such as potency assays, regulatory review, and quality control, are time-limiting processes where efficiencies need to be realized through collaborative approaches. It is also essential to develop alternative supply chain sources or vaccine manufacturing processes that create less time dependence and de-risk the time to respond. Endeavours such as the Developing Countries Vaccine Manufacturers Network (DCVMN), established in the year 2000 and currently encompassing 41 manufacturers in 14 countries, enhance global capability and reduce dependency on developed nations [24]. Other efforts include the WHO Global Pandemic Influenza Action Plan (GAP), of which one effort initiated in 2006 was aimed at increasing the world’s production capacity for pandemic vaccine by supporting six developing country manufacturers in the production of influenza vaccines, resulting in limited success [25]. Lack of access and distribution difficulties with COVID-19 vaccines to developing countries has been highlighted in the response to the current pandemic [26]. The WHO continue to expand their network of surveillance laboratories, and with their partners have a continuous improvement program to expand sequence capability in their national influenza centres through next generation sequencing to facilitate the identification of novel viruses.

## 3. A(H1N1) 2009 Pandemic Response Timelines as a Model

A novel influenza A (H1N1) outbreak had begun to emerge in March 2009 in Mexico with notable increases in hospitalizations [27]. On 15 April 2009, a swine-lineage A(H1N1) influenza strain was identified in a paediatric patient in California, which was later defined as the first case (Figure 1). By the end of April, cases had appeared throughout North America and in Europe [17,18,28]. By mid-May, half of all influenza samples tested were identified as the new H1N1 strain, and by the end of May, cases had spread throughout the globe [17,18,29,30,31]. On 11 June 2009, the WHO reported 30,000 confirmed infections in 74 countries and declared a pandemic [17,18].

Within a few days of sequence identification of the novel, pandemic virus strain, the CDC and other members of the WHO Global Influenza Surveillance Network (now GISRS) increased surveillance and began investigating potential candidate vaccine viruses (CVV), with a focus on finding virus strains that could be propagated at high yield in fertilized eggs (the predominant manufacturing platform of the time). Vaccine trials began in mid-July, less than 100 days from detection of the first case. Trials were completed, vaccines licensed, and production underway less than 100 days from the pandemic declaration. Vaccine distribution began in the US on 30 September, 11 days after the 100-day milestone, and the first dose was administered 5 days later [17]. However, the first peak of cases occurred prior to substantial roll out of the A(H1N1) 2009 pandemic vaccine. The timelines for availability of a pandemic vaccine could have been further shortened if existing seasonal influenza vaccine data were used as a guide for safety, assuming the dosage was similar.

Although CVV investigations began shortly after the index case, several challenges had to be overcome before clinical trials could proceed. First, CVVs had to be generated through two distinct methods: either reassortment with high-yield laboratory strains or the synthesis of high-yield pandemic strains by reverse genetic synthesis within plasmids. The initial reverse genetic CVVs produced in qualified mammalian cells did not produce sufficient cell titres in embryonated chicken eggs, and thus only viruses developed by traditional methods were utilized. Second, the wild-type A(H1N1) 2009 virus was classified as a Biosafety Level 3 (BSL-3) virus, and the CVV development had to be conducted within BSL-3 facilities. Finally, the safety of attenuated CVVs had to be demonstrated in animal studies in BSL-3 facilities. By late May, the first CVVs were made available to manufacturers, and human trials to determine the vaccine dose and regimen began roughly 7 weeks later in July [17,31]. For example, in Australia, clinical trials were identified as being needed and the first dose was administered on 22 July 2009, 62 days after the WHO pandemic declaration [32].

In 2009 and to this day, influenza vaccines are formulated based on potency assays, a process that takes several months due to the requirement to develop and calibrate a specific antiserum raised against the CVV and reference antigens to be used in a single radial immunodiffusion (SRID) assay—the only assay currently accepted by regulators for this purpose [31,33].

The US Food and Drug Administration (FDA) authorized four injectable (CSL Limited, ID Biomedical Corporation, Novartis Vaccines and Diagnostics Limited, Sanofi Pasteur, Inc.,) and one intranasal (MedImmune LLC) monovalent vaccines to combat A(H1N1) 2009 on 15 September, 97 days after pandemic declaration, and the European Medicines Agency (EMA) authorized three vaccines (Nanotherapeutics Bohumil, Novartis Vaccines and Diagnostics Limited, and GlaxoSmithKline) on 29 September, 111 days after pandemic declaration, followed by two more pandemic vaccines in March and June 2010 [29,30]. A key factor in the swift approval of these vaccines was the decision by the Vaccines and Related Biological Products Advisory Committee (VRBPAC) and the EMA to endorse licensure based on the *strain change* pathway (US) or “*mock-up*” vaccine approach (EU). These decisions permitted the pandemic vaccines to be produced as a monovalent vaccine containing antigens for A(H1N1) 2009 using existing FDA or EMA-approved platforms and procedures for the manufacture of seasonal and pandemic influenza vaccines [17,30]. The requirement for additional immunogenicity or safety data depended on the region or nation’s regulatory framework and the pre-existing availability of virus strain clinical data. Neither immunogenicity nor new safety data were required for the initial authorization in the US (except for safety confirmation of the live attenuated vaccine used in the nasal formulation), as the 2009 H1N1 vaccine was made in the same way as standards already in place for seasonal vaccines [17,29]. In the EU, clinical trial data were provided as they became available in a rolling review process that did not delay authorization of the pandemic vaccines [30].

Vaccine distribution began in October 2009 with initial demand for the vaccine exceeding the available manufacturing capacity at that time [17,34,35,36]. In some places, vaccine roll-out was prioritized to the most vulnerable populations but supply and distribution were not always uniform [24,25]. A 10-year retrospective analysis of the barriers identified in 2009 and the progress in technical development since suggests that the world is better positioned to respond to an influenza pandemic than it was in 2009 [31], but the timely supply of influenza vaccine to meet global needs in an influenza pandemic would still be a challenge.

## 4. Improvements in Influenza Pandemic Preparedness and Response since 2009

The WHO GISRS continues to expand, with more countries performing influenza surveillance, including activity at the animal-human interface, which is critical for early detection of emerging IVPPs that may lead to outbreaks and pandemics. However, this enhanced surveillance and subsequent response may potentially be affected by other international agreements, such as the Nagoya Protocol on Access to Genetic Resources and the Fair and Equitable Sharing of Benefits Arising from the Utilization (ABS), which is a supplementary agreement to the Convention on Biological Diversity. This agreement, which came into force in 2014, and currently includes 133 countries that are Parties to the Protocol, may complicate the sharing of influenza virus isolates and slow vaccine manufacture due to its requirement for bilateral agreements between countries and individual manufacturers, despite the agreement being mindful of the importance of ensuring access to human pathogens for public health preparedness and response [37].

The WHO also works with the World Organization for Animal Health (OIE) and the Food and Agriculture Organization of the United Nations (FAO) in a “One Health” approach to influenza surveillance. Information on influenza viruses circulating in animals is a key element in identifying influenza viruses with the potential to cause infection and disease in humans.

Over the past decade, continuous improvements have also been made in influenza science, manufacturing processes, production capacity, and regulatory processes which should help shorten the response time to future influenza pandemics. Advances in reverse genetic methodologies have shortened the timeframe for CVV development and testing by 2–3 weeks, and the emergence of cell-based viral propagation and recombinant viral protein technologies should also lessen timelines [31]. Ongoing optimization of seasonal influenza vaccine production, including improvements that, for example, facilitate a better antigenic match between vaccine and circulating virus strains, contribute to process improvements that will inform and support pandemic vaccine efforts. Each seasonal influenza vaccine improvement may be carried over into the mock-up license or strain change vaccines that the EMA, FDA, and other agencies may authorize through a fast-track process in the event of a new influenza pandemic [34,38]. Some of these advances have lessened the reliance on egg propagation of seasonal vaccine viruses, although currently the majority of the vaccine manufacturing is still egg-based [39]. In 2021, 82% of US seasonal influenza vaccines contained antigens from viruses propagated in eggs [40], and this proportion is higher in other countries.

Advancements in methods to assess the potency of influenza vaccines have also been made in recent years. The SRID assay for the determination of the content of hemagglutinin in inactivated influenza vaccines was developed in the 1970s and remains the only accepted assay to determine the potency of vaccine formulations nearly 50 years later [33]. Alternate approaches that significantly reduce timeframes include enzyme-linked immunosorbent assay (ELISA), high-performance liquid chromatography (HPLC), mass spectroscopy, and other approaches that have the potential for rapid yet accurate results, in some cases without the need for specific reagents [31,41,42,43]. While these newer methods have been widely utilized, they have yet to be introduced routinely into the influenza vaccine production processes or to be used for regulatory approval of vaccines, a situation that urgently needs resolution [44].

In addition to fast-track procedures which include the authorization of mock-up files or strain changes for vaccines against pandemic threats [34,38], governments can stockpile zoonotic influenza antigens from influenza viruses that pose a pandemic threat in bulk preparations, which can be rapidly adapted into vaccine doses and used against outbreaks of the same viral lineage or subtype until vaccines specific to the pandemic strain are developed.

Another key element is the role of adjuvants, which may be used in authorized pandemic vaccines to improve the efficacy or possibly to reduce the dose—a notable component of research and development efforts and an important consideration in manufacturing scale-up. Adjuvants are commonly used in vaccines to boost the immune response and prolong or enhance antigen exposure [45]. Adjuvants can also provide an antigen-sparing effect, which means less antigen is needed within each vaccine dose, and thus more vaccine doses can be made from the available antigen supply—a vital step in ensuring that vaccine production meets pandemic demands [31]. Advances in the use of adjuvants in influenza vaccines have recently been reviewed [46].

Through the Global Action Plan for Influenza Vaccines (2006–2016), the WHO promoted an increase in vaccine production capacity, such that the potential annual global pandemic vaccine capacity could be increased to 8.31 billion doses by 2015—a figure that as of 2019 was considered only possible in the best-case scenario [31,47,48]. Collaboration between government and industry has also been strengthened since both the A(H1N1) 2009 and the COVID-19 pandemic. In the US, the National Influenza Vaccine Modernization Strategy (NIVMS) 2020–2030 is an extensive plan to reinforce and diversify influenza vaccine development, manufacturing, and the supply chain by partnering with both domestic and international vaccine manufacturers [39]. In alignment with this strategy, the Biomedical Advanced Research and Development Authority (BARDA) has partnered with multiple influenza vaccine manufacturers to expand manufacturing capacity, streamline preparation of CVVs, and improve the antigen yield in eggs and mammalian cell lines. BARDA also supports research into alternative potency assays and adoption of rapid sterility assays for vaccine manufacturing to facilitate more rapid availability of seasonal and pandemic influenza vaccines [49].

Finally, national and international organizations have renewed commitments to information sharing with viral surveillance networks and directed additional resources into these agencies to monitor viral outbreaks throughout the world. For example, in addition to the WHO GISRS and US IRAT, the UK New and Emerging Respiratory Virus Threats Group (NERVTAG) was established in 2014 to assess scientific risk and advise on mitigation and management strategies for new and emerging respiratory virus threats [50] and the European Commission is launching the European Health Emergency preparedness and Response Authority (HERA) to prevent, detect and rapidly respond to future health emergencies [51].

## 5. New and Emerging Vaccine Technologies

One of the strategies for mitigating future pandemics is to leverage the new vaccine technologies used for COVID-19 vaccines, alongside existing influenza vaccine platforms. Table 1 summarizes these approaches and their current development status for influenza vaccines [1,48,52,53,54,55].

The success of new platforms such as mRNA and viral vector vaccines against COVID-19 have generated much excitement over the possible applications of these technologies to other diseases [5,56,57]. COVID-19 mRNA vaccines use genetically engineered mRNA protected and transported within a lipid nanoparticle to give instructions to the recipient cell to make spike (S)-protein, whereas viral vector vaccines that have been used to date use encode the S-protein gene from the COVID-19 virus into various adenoviruses. This vector delivers the viral genetic material into the cell to direct it to make copies of the encoded S-protein but does not lead to the replication of new infectious adenoviruses.

New vaccine technologies may shorten initial vaccine synthesis timeframes, for example the core of the mRNA vaccine can be synthesized from a DNA template within a few hours, compared with several days to synthesize influenza virus antigens in eggs or mammalian cell lines. The remaining steps, however, including regulatory procedures, manufacturing, and product release, are not necessarily faster than those for existing influenza vaccines. Vaccine manufacture involves a series of complex and sequenced steps with many commonalities across the vaccine technologies, for both novel and established platforms, but also some unique challenges (Table 2) [58,59,60,61]. For example, the formulation, isolation, purification, and storage and shipping of mRNA vaccines remain complex. The mRNA synthesis and capping process requires the use of a DNA template, an RNA polymerase, and modified nucleotide triphosphate (NTP) substrates, and other specialized enzymes and other components. These must be obtained from certified suppliers that can guarantee each component meets good manufacturing practice (GMP) standards. As a result, supplies may be limited and costs high. Purification is also limited by a lack of cost-effective pharmaceutical-scale chromatographic methods able to remove residual NTPs, double-stranded RNA, template DNA fragments, and other impurities without the use of toxic reagents and/or complex, multistep purification processes. Finally, the current extreme cold storage or transport requirements of some mRNA vaccines may make them challenging for distribution to some parts of the world—including more remote areas of developed countries [56].

More broadly, a key question for the use of novel vaccine technologies against seasonal influenza will be if they have an acceptable reactogenicity and risk–benefit profile if they need to be used annually across a wide range of age groups. For example, the uncommon adverse events such as cardiac and haematological side effects associated with the mRNA and adenovirus COVID-19 vaccines, respectively, are yet to be fully understood and mitigated. For some vaccines, such as those based on an adenovirus vector or other viral vectors, there is also the added complication of host responses to the adenovirus itself, which may limit the frequency at which the vector can be used.

Currently, development and regulatory timelines, complexities of scaling the isolation and purification steps, and typical supply chain requirements mean that mRNA and/or adenovirus vaccines may have similar challenges to existing platforms and thus are not necessarily a faster option than influenza vaccines manufactured using currently established methods. However, these approaches represent an important addition to the means of combatting future influenza pandemics and possibly seasonal influenza, but further work and clinical trials will be required.

## 6. Conclusions

The availability of vaccines for pandemics is complex and depends on numerous factors beyond the science of identifying the pandemic threat and generating a vaccine. For the optimal response, all stakeholders, including governments and other agencies, must openly share information and cooperate with each other to control the spread of the pandemic. Vaccine innovations provide important tools to help combat the next pandemic, and a variety of platforms, including established as well as newer technologies, will help maintain a state of readiness through the foreseeable future. Building on existing manufacturing systems as new approaches come online will also ensure global capacity for widespread vaccinations.

Annual production of seasonal influenza vaccine is a complex process that involves the management of many variables, including antigenic variability, multivalent vaccines, and ongoing efforts to increase vaccine effectiveness and manufacturing speed. These complexities are in addition to the scientific challenges inherent to influenza viral evolution.

The influenza vaccine community has a long history of meeting these challenges each influenza season, year after year, and this experience has shaped the framework for rapid development of pandemic influenza vaccines based on established technologies. Active and sustained manufacturing of seasonal influenza vaccines provides a wealth of data on the safety, efficacy, and effectiveness of vaccines and adjuvants, as well as existing operational platforms that can be rapidly repurposed to address future influenza pandemic needs. This experience in the influenza vaccine community will undoubtedly be invaluable in the next influenza pandemic.

Influenza poses one of the greatest risks for the next pandemic. Countering it will require numerous innovations, many of which can be built upon the knowledge and experience of past influenza pandemics and public health responses to the pandemics. Existing influenza pandemic vaccines are important in their own right as a public health intervention, but they also provide important lessons and a blueprint for the development of novel technologies that can assist in the mitigation of future influenza pandemics.

## Figures and Tables

**Figure 1 vaccines-10-00589-f001:**
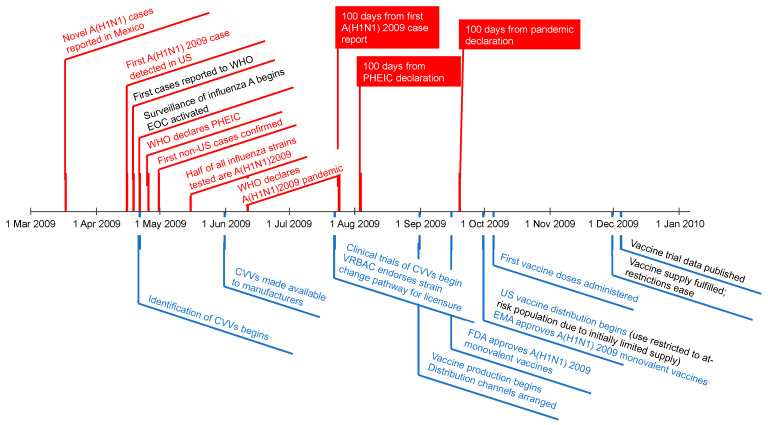
Timeline of A(H1N1) 2009 pandemic. Pandemic-related events are in red type; vaccine development–related events are in blue type [17,18,27,29,30,31]. ACIP, US Advisory Committee on Immunization Practices; CDC, US Centers for Disease Control and Prevention; CVV, candidate vaccine viruses; EMA, European Medicines Agency; EOC, CDC Emergency Operations Center; PHEIC, public health emergency of international concern; VRBAC, Vaccines and Related Biological Advisory Committee; WHO, World Health Organization.

**Table 1 vaccines-10-00589-t001:** Established, new, and emerging vaccine technologies [1,52,53,54,55].

Vaccine Platform	COVID-19 Vaccine Status *	Influenza Vaccine Status *
**Established**		
Egg-based, inactivated split virion or protein subunit	None in human trials	Licensed seasonal
Egg based inactivated split virion or protein subunit, adjuvanted	None in human trials	Licensed pandemic and seasonal
Egg-based live attenuated	None in human trials	Licensed seasonal
Cell culture–based, purified protein subunit	Licensed	Licensed seasonal
Cell culture–based, purified protein subunit, adjuvanted	Licensed	Licensed pandemic
r-Protein subunit	Licensed	Licensed pandemic and seasonal
**New/emerging**		
mRNA	Licensed	Phase 1/2
sa-mRNA	Phase 1	Preclinical
Viral vector	Licensed	Preclinical
Combination (influenza/COVID)	COVID + seasonal influenza in phase 1/2	
DNA	Licensed	None in human trials

* Licensed refers to approval in major regions and may include full licensure and/or emergency/conditional authorizations. This summary is based on information available at the time of manuscript submission and represents the latest phase of a vaccine in major markets/regions and is therefore subject to change. There may be other vaccines that are in earlier stages of development, and some of these categories or classifications may overlap. For accuracy and latest updates, please refer to the relevant marketing authorizations in respective regions. For a list of COVID-19 vaccines status and manufacturers, please refer to the WHO tracker [1]. A list of influenza vaccine manufacturers from a recent review [48] appears in Appendix A.

**Table 2 vaccines-10-00589-t002:** Key manufacturing steps for selected vaccine platforms *.

Platform	Source and Prepare (after Strain Selection)	Propagation, Harvest, and Inactivation	Splitting, Purification, and Filtration	Bulk Production	In-Process and DS Testing	Formulation and Filtration	Filling	Inspection, Labelling, and Packaging	Final Product Release Testing and QA Review
IIV and purified surface antigen	WHO provides wild-type virus to the reassortment laboratories. CVV made available to manufacturer Synthetic seed prepared using genetic sequence shared on publicly accessible database (e.g., GISAID)	Incubate in hens’ eggs or mammalian cells for virus replication Harvest fluid containing virus Virus inactivation (some processes position virus inactivation just before final filtration)	Splitting/disruption of virus depending on specific vaccine (except for whole virion vaccine) Bulk antigen purification (ultra centrifugation on saccharose gradient, filtration, or alternate separation steps	Sterile filtration	QC DS release testing including: Potency, sterility, purity and impurities	Dilution and sterile filtration Mix with adjuvant (if applicable) QC	Filling into vials, syringes or other administration form (e.g., sprayers)	Automated, semi-automated, or manual visual inspection of the filled material Labelling and packaging (country or region specific)	Internal QC product release assays, including: Potency, sterility, purity and impurities For bulk, formulated bulk, and fill and pack steps: Deviation investigation, QA review, and closure Internal manufacturing and QA batch dossier review and final release Additional packaging, as required Submission of BPR to external regulatory agency(ies). External regulatory agency(ies)’ testing and release of product
Live attenuated influenza virus	Genetic sequence provided by WHO or GISAID (wild-type viruses for IVPP not usually shipped) Manufacturer initiates virus reassortment by reverse genetics; propagated in eggs to produce CVV	Incubate in hens’ eggs for virus to replicate Harvest fluid containing virus	Clarification and concentration Sterile filtration (if possible) Freezing		QC DS release testing including: Potency, bioburden, sterility, purity and impurities	Dilution and sterile filtration QC			
r-Protein subunit	Combine gene with baculovirus to make recombinant HA Plasmid construction	Engineering cell expression: inoculate cultured mammalian cells to replicate HA	Clarification, centrifugation, chromatography	Bulk antigen is sterile filtered, collected, and frozen	QC Potency, sterility, purity and impurities	Mix with adjuvant (if applicable) or extemporaneous addition of adjuvant			
Viral vector vaccine	Genomic sequence Cell banks and virus seed stocks	Cell culture Transfection (into viral DNA into cells) Virus infection, viral vector production	Virus propagation Viral vector purification Ultracentrifugation, chromatography, purification solutions		QC Potency, sterility, purity and impurities	Ultrafiltration Viral vector QC			
mRNA vaccine	Manufacture DNA template, insert into plasmid DNA In vitro transcription	Transcribe mRNA Degrade by DNase step Addition of the cap	High pressure LC Chromatography, adjust concentration, filtration, freezing		QC Potency, sterility	LNP formulation Sterile filtration QC			

* Manufacturing steps do not include all supporting activities, such as manufacture of adjuvant (where relevant), manufacture of buffers, and any reagent preparation [58,59,60,61]. BPR, batch release protocol; CVV, candidate vaccine virus; DS, drug substance; GISAID, Global Initiative on Sharing Avian Influenza Data; IIV, inactivated influenza virus; HA, hemagglutinin; IVPP, influenza viruses of pandemic potential; LAIV, live attenuated influenza virus; LC, liquid chromatography; LNP, liquid nanoparticle; QA, quality assurance; QC, quality control; WHO, World Health Organization.

## Data Availability

Not applicable.

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
