# Peer review of "Global Pandemic Preparedness: Optimizing Our Capabilities and the Influenza Experience"

_vaccines, 2022, doi:10.3390/vaccines10040589_

Round 1

Reviewer 1 Report

Rockman et al have discussed the implications of the 2009 H1N1 and COVID-19 pandemic on the preparedness for future influenza pandemics. They comprehensibly discuss various government and regulatory frameworks that are in place for pandemic preparedness. However further connections could be made in how these frameworks where leveraged during the COVID-19 pandemic, and how the 2009 H1N1 pandemic instigated these updates. Specific discussions on the implications for future pandemics where improvements have been made could be discussed, i.e. next generation vaccines in clinical trials, newly approved adjuvants, adaptability of mRNA and adenoviral vaccines for plug and play formats and the scale of their production beyond the USA within 100 days.

Comments to the author

  1. The authors email and affiliation is mixed and incomplete (first and last author emails given, are they both corresponding authors?).
  2. The abstract mentions HIV as an exclusion to influenza pandemics (and is never mentioned again in the main text) and opening introduction mentions influenza contributing to 4 of 5 pandemics, this could be restructured for context.
  3. “The first mRNA vaccines were investigated in the late 1980s, and adenovirus vector vaccines have also been studied for many years” -> how many years? Specify like mRNA vaccines their first appearance in pre-clinical or clinical trials.
  4. Line 55-77, much of the influenza reporting platforms where adopted to SARS-CoV-2 reporting, including sequence deposition platforms GISAID/GISRS and nextseq etc- without these influenza tools in place the COVID-19 pandemic response would have been hindered. The discussion of the COVID-19 pandemic could be more integrated rather than versus influenza, these platforms made a synergistic accelerated response that would not have been possible without H5N1/SARS/MERS/H1N1 pandemic warning signals.
  5. Line 124- The 100-day goal of a vaccine, what scale would it be available at outside of the USA?
  6. Line 126-130, the use of non-injectable and cold chain independent stable vaccines for easier distribution is seperate to the speed of mRNA and Adenoviral vector vaccine production- what specific efforts are being made on both fronts? These issues are two-fold, speed and distribution, the discussion could be expanded here.
  7. Have GAP and DCVMN been active in the production of COVID-19 vaccines? Are they operational now for influenza pandemic vaccines? Have these infrastructures been realized?
  8. Vaccine hesitancy should also be discussed for novel rapid vaccines which has hindered the public health response in some countries considerably.
  9. Figure 1 is an excellent resource for the 2009 pandemic timeline, what do Rockman et al forsee as the timeline in a future influenza pandemic given the developments in manufacturing technology since 2009 and the COVID19 pandemic? Which stages have been augmented for future pandemics and which areas remain hurdles (potency testing)? The figure should also highlight that the H1N1 2009 vaccine was available AFTER the peak of cases, so whilst the 100 day timeline is laudable it was still too late.
  10. Line 151-159- The time lag of the cluster of cases in Mexico in March 2009 and then reporting in the USA in mid-April 2009 due to sequencing difficulty to identify the new H1N1 strain, may have delayed containment and the initial response. This is reminiscent of the emergence in Wuhan of SARS-CoV-2 where local labs and the upwards reporting delayed identification of the new viral strain from the small 27 case cluster. What changes have been made in the identification and reporting of new influenza viruses since the 2009 pandemic? Has diagnostic primers and sequencing capacity or virus isolation increased in more regional areas to accelerate the speed of identification?
  11. Line 192, describe the potency and SIRD assay? Have changes been made to these protocols since 1975 (first described) or 2009 (updated since pandemic)? Reference serum is ferret based? Further details on this key hurdle for regulatory approval are needed. The later discussion on ELISA and other approaches and BARDAs investment are a strength, but the actual assay could be briefly described on why it is critical for vaccine approval.
  12. Line 224, what would the specific ‘struggle’ be for the distribution of influenza vaccines in a future pandemic?
  13. What do the Nagoya and ABS agreements mean for the country which identifies a new strain used for vaccine manufacturing elsewhere? The ‘mindful’ language seems ambiguous in the case of a pandemic, will this hinder future responses?
  14. Line 270, the stockpile of H5N1 and H7N9 vaccines in the USA should be described, the scale, cost and replenishment of stocks, and clinical trials of these vaccines showing poor immunogenicity needing adjuvants in combination…hence the need for new platforms such as mRNA, cell based, adenoviral vectors…
  15. Line 276, the use of adjuvants for antigen sparing in the 2009 monovalent vaccine was associated but disputed with Narcolepsy in Sweden and may have an altered safety profile. Also, what new adjuvants have been licensed since the 2009 pandemic? Are any approved for the use in influenza vaccines? These specifics could be discussed.
  16. Line 331, typo ‘howeverincluding’
  17. Figure 2 is a table not a figure, the text is difficult to read, formatting like Table 1 in landscape view would help readability.

Author Response

Uploaded as a WORD file.

Reviewer 2 Report

Rockman and colleagues review the pandemic preparedness for influenza, the new ambition to develop new pandemic vaccines in under 100 days, review the technologies available, particularly mRNA vaccines and  identify improvements (surveillance, production, potency testing, adjuvants). Interestingly, they conclude that the conventional method to make influenza vaccines should not be underestimated. Indeed the show that in 2009 the old egg based platform to make vaccines met the ambition of CEPI and WHO to get vaccine approval in 100 days. This is because for this platform, safety, is known, regulatory pathways are well established and manufacturing capacity is ready to go since it is kept worn by the annual manufacturing of seasonal vaccine. Overall, there is no much novelty in this review,  and describes well known things, however in this moment where the global attention is only focused on innovative but less established technologies, it is important to publish this paper to remind people to not forget this less popular but effective technology.

I suggest to publish as it is. The only suggestion for the authors is to consider to add names of the Flu vaccines in Table 1 and include additional information on preclinical and clinical studies of new technologies for flu vaccines to enhance the significance of the manuscript.

Author Response

Uploaded as a WORD file.

Reviewer 3 Report

Manuscript by Steven Rockman et. al. entitled “Global Pandemic Preparedness: Optimizing Our Capabilities and the Influenza Experience” is an opinion type of article describing several aspects of vaccine manufacturing strategies as well as improvement and changes made recently in influenza vaccine development and key factors for successful, “in-time” vaccine. Authors include comparisons of various vaccine strategies and discuss impact of COVID-19 pandemic on influenza vaccine development. The manuscript is very interesting and well-written. It brings the attention to influenza virus which has pandemic potential and condense the knowledge about current trends in vaccines platforms. I recommend to accept a manuscript after small improvement.

In Table 1 please add a firms that developed and testing the listed vaccine technology and vaccines.

I also would like to ask authors to check if the information are up-to-date about the vaccine trials status in Table 1.

Author Response

Uploaded as a WORD file.
